# Agreement between upper and lower limb measures to identify older adults with low skeletal muscle strength, muscle mass and muscle quality

**Charles Phillipe de Lucena Alves**[1], **Marcyo Câmara** [2], **Geovani Araújo Dantas Macêdo**[1], **Yuri Alberto Freire**[2], **Raíssa de Melo Silva**[1], **Ronildo Paulo-Pereira** [2], **Luiz Fernando Farias-Junior**[3], **Ana Paula Trussardi Fayh**[1,2,4], **Arnaldo Luis Mortatti**[1], **Eduardo Caldas Costa**[1,2]*

1 Graduate Program in Physical Education, Federal University of Rio Grande do Norte, Natal, RN, Brazil,
2 Graduate Program in Health Sciences, Federal University of Rio Grande do Norte, Natal, RN, Brazil,
3 Graduate Programa in Psychobiology, Federal University of Rio Grande do Norte, Natal, RN, Brazil,
4 Graduate Program in Nutrition, Federal University of Rio Grande do Norte, Natal, RN, Brazil

* ecc@ufrnet.br

## Abstract

### Background

Identifying low skeletal muscle strength (SMS), skeletal muscle mass (SMM) and skeletal muscle quality (SMQ) is pivotal for diagnosing sarcopenia cases. Age-related declines in SMS, SMM, and SMQ are dissimilar between the upper (UL) and lower limbs (LL). Despite this, both UL and LL measures have been used to assess SMS, SMM and SMQ in older adults. However, it is not clear whether there is agreement between UL and LL measures to identify older adults with low SMS, SMM and SMQ.

### Objective

To investigate the agreement between UL and LL measures to identify older adults with low SMS, SMM and SMQ.

### Methods

Participants (n = 385; 66.1 ± 5.1 years; 75,4% females) performed the handgrip strength test (HGS) and the 30-s chair stand test (CST) to assess UL- and LL-SMS, respectively. The SMM was assessed by dual-energy X-ray absorptiometry (DXA). The UL-SMQ was determined as: handgrip strength (kgf) ÷ arm SMM (kg). LL-SMQ was determined as: 30-s CST performance (repetitions) ÷ leg SMM (kg). Results below the 25th percentile stratified by sex and age group (60–69 and 70–80 years) were used to determine low SMS, SMM and SMQ. Cohen's kappa coefficient (κ) was used for the agreement analyses.

**Data Availability Statement:** All files are available from the Open Science Framework at: https://osf.io/suavh/ (DOI 10.17605/OSF.IO/SUAVH).

**Funding:** The author(s) received no specific funding for this work.

**Competing interests:** The authors have declared that no competing interests exist.

## Results

There was a slight and non-significant agreement between UL and LL measures to identify older adults with low SMS (κ = 0.046; 95% CI 0.093–0.185; p = 0.352). There was a moderate agreement to identify low SMM (κ = 0.473; 95% CI 0.371–0.574; p = 0.001) and a fair agreement to identify low SMQ (κ = 0.206; 95% CI 0.082 to 0.330; p = 0.005).

## Conclusion

The agreement between UL and LL measures to identify older adults with low SMS, SMM and SMQ is limited, which might generate different clinical interpretations for diagnosing sarcopenia cases.

## 1 Introduction

Aging is commonly accompanied by declines in skeletal muscle strength (SMS), skeletal muscle mass (SMM) and skeletal muscle quality (SMQ) [1–4]. SMQ can be defined in terms of muscle composition or relative strength [5, 6]. SMQ (as relative strength) describes the muscle's ability to function and is operationally defined in terms of SMS normalized to SMM [5, 6]. Assessing SMS, SMM, and SMQ in older adults has been recommended by clinical guidelines. For example, the European Working Group on Sarcopenia in Older People (EWGSOP2) recommends assessing SMS, SMM, and SMQ to identify those who are at high risk for or have established sarcopenia [7]. In addition, low SMS, SMM, and SMQ are associated with a higher risk for several adverse health-related outcomes in older adults, such as reduced mobility [8], physical disability [8], frailty [9], falls [10], impaired health-related quality of life [11, 12], all-cause and cardiovascular mortality [13–16].

Age-related declines in SMS, SMM, and SMQ are dissimilar between the upper (UL) and lower limbs (LL) [1–4]. Despite this, both UL and LL measures have been used to assess SMS, SMM and SMQ in older adults [17–20]. The handgrip strength test (HGS) and the chair stand test (CST) have been commonly used in clinical practice to assess SMS in older adults. However, a previous study [20] demonstrated that the prevalence of older adults at high risk for (low SMS) and having established sarcopenia (low SMS + low SMM) was lower using HGS than the 5-repetition CST. In addition, the authors observed poor agreement between the HGS and the 5-repetition CST to identify both individuals at high risk for and having established sarcopenia, suggesting that the interchangeable use of these tests might generate different clinical interpretations for the EWGSOP2 algorithm [7]. Thus, more information is needed about the agreement between UL and LL measures to identify older adults with impaired neuromuscular characteristics. In view of this, the aim of this study was to investigate the agreement between UL and LL measures to identify older adults with low SMS, SMM and SMQ.

## 2 Methods

### 2.1 Study design

This was a cross-sectional study which is reported in accordance with the STROBE (STrengthening the Reporting of OBservational studies in Epidemiology) statement [21]. This study was conducted at the Onofre Lopes University Hospital and at the Department of Physical

Education of the Federal University of Rio Grande do Norte between June 2018 and December 2019. The Ethics Committee of the Federal University of Rio Grande do Norte approved this study (protocol number: 2.603.422/2018), which was conducted according to the Declaration of Helsinki. All participants were informed about the study procedures and gave written informed consent.

## 2.2 Participants

Community-dwelling older adults aged 60–80 years from the city of Natal, RN, Brazil were recruited to participate in this study by advertisements on radio, TV, e-flyers in social media sites, healthcare units, and older adult community centers. The inclusion criteria were: i) no history of known cardiovascular diseases or major cardiovascular events (e.g., acute myocardial infarction, stroke, coronary artery disease, arrhythmias, or peripheral vascular disease); ii) no muscle, joint or bone injury which limits the ability to perform exercise; iii) no acute diabetes- or hypertension-related decompensation (i.e. glycaemia $\geq$ 300 mg/dL; blood pressure $\geq$ 160/105 mmHg). Participants with incomplete data related to the strength tests or body composition assessment were excluded from the final analysis.

## 2.3 Skeletal muscle strength

**2.3.1 Handgrip strength test.**   The HGS was performed following the recommendations of Coldham [22] as a proxy of UL-SMS. All participants were seated in a straight-backed chair with their feet flat on the floor and positioned in a standardized position with their shoulder adducted and neutrally rotated, elbow flexed at 90˚, forearm in a neutral rotation, and their wrist between 0˚ and 30˚ extension and between 0˚ and 15˚ ulnar deviation. All participants were instructed to squeeze the handgrip (Jamar® 5030J1) as hard as possible during a 5-second period with their dominant hand during the expiration phase, avoiding Valsalva's maneuver. They performed three attempts with verbal encouragement interspersed by 1-minute interval between each attempt. The highest value observed in the three attempts was considered for data analysis.

**2.3.2 30-s chair stand test.**   The 30-s CST was performed following the recommendations of Rikli and Jones [23] as a proxy of LL-SMS. The participants were instructed to sit in the middle of the chair with their back straight, feet flat on the floor, and arms crossed at the wrists and held against their chest. On the signal "go", they were instructed to rise to a full stand and then return to a fully seated position. All participants were verbally encouraged to complete as many full stands as possible within a 30-s period. The number of repetitions was considered for data analysis.

## 2.4 Skeletal muscle mass

Dual-energy X-ray absorptiometry (DXA) is a widely used technique which assesses body composition at the molecular level [24, 25]. It assesses the lean soft tissue (LST) or lean body mass, which is the sum of body water, total body protein, carbohydrates, non-fat lipids, and soft tissue mineral [24, 25]. Body composition was assessed by DXA (GE Healthcare® Lunar Prodigy Advance) following the recommendations of the National Health and Nutrition Examination Survey [26]. Participants' weight (kg) and height (cm) were previously measured (Welmy® W300). Total-, arm- and leg-LST in kilograms were calculated by specific software (Encore, version 14.1) from the DXA scan. In this study, LST determined by the DXA technique was used as a proxy of total-, arm-, and leg-SMM [24], as recommended by the European Society for Clinical and Economic Aspects of Osteoporosis and Osteoarthritis working group on frailty and sarcopenia.

## 2.5 Muscle quality

SMQ was determined in terms of UL- and LL-SMS (HGS and 30-s CST, respectively) normalized to appendicular skeletal muscle mass (ASM; kg) as assessed by DXA [5, 6]. Therefore, UL-SMQ was determined as: HGS (kgf) ÷ arm SMM (kg). LL-SMQ was determined as: 30-s CST performance (repetitions) ÷ leg SMM (kg).

## 2.6 Criteria for defining older adults with low SMS, SMM and SMQ

The UL and LL measures for SMS, SMM and SMQ were stratified into quartiles based on sex and age group (60–69 and 70–80 years). Males and females from each sex and age group who had UL and LL measures for SMS, SMM and SMQ below the 25th percentile were identified as older adults with low SMS, SMM and SMQ [23, 27].

## 2.7 Physical activity

Physical activity level was determined by the Brazilian version of the Minnesota Leisure Time Activities Questionnaire for older adults [28]. The physical activities were classified as light, moderate or vigorous considering the absolute intensity (metabolic equivalents; METs) for each specific age (40–64 years; $\geq$ 65 years), based on the American College of Sports Medicine [29]. Participants who performed $\geq$600 MET/min/wk of moderate-vigorous physical activities were considered as 'active', while those who performed $<$ 600 MET/min/wk were considered as 'inactive'.

## 2.8 Statistical analysis

Descriptive data are expressed as mean ± standard deviation, absolute and relative frequencies. Data normality was verified by Shapiro-Wilk and Q-Q plot tests. Cohen's kappa coefficient ($\kappa$) was used to analyze the agreement between UL and LL measures to identify older adults with low SMS, SMM and SMQ. Cohen's kappa coefficient ($\kappa$) < 0.00 was interpreted as poor agreement, 0.00–0.20 as slight agreement, 0.21–0.40 as fair agreement, 0.41–0.60 as moderate agreement, 0.61–0.80 as substantial agreement, and 0.81–1.00 as almost perfect [30]. The significance level was set at $p < 0.05$ for all analyses.

# 3 Results

A total of 385 older adults were included in the final analysis (Fig 1). Most participants were females (72.4%; n = 279), Caucasian (42%; n = 163) and 'Pardos' or Brown (49.7%; n = 193), lived with a partner (69.1%; n = 266), were overweight or obese (overweight: 40.0%, n = 154; obesity: 38.4%, n = 148), and had hypertension (52.5%; n = 202). Approximately one-third of the participants were ex-smokers (35.3%; n = 136) and had dyslipidemia (32.9%; n = 127). Few participants had post-secondary education (4.2%; n = 16), were smokers (3.4%; n = 13), or had diabetes (14.3%; n = 54). Additionally, 58.4% (n = 225) were physically active and 41.6% (n = 160) were physically inactive. Table 1 shows the neuromuscular characteristics of the participants.

Table 2 shows the cut-offs (25th percentile) to identify older adults with low SMS, SMM and SMQ, according to sex and age group. Overall, the cut-offs for the neuromuscular characteristics were slightly lower for females and older adults aged 70–80 years.

Table 3 shows the agreement analysis between UL and LL measures to identify older adults with low SMS, SMM, and SMQ. There was a slight and non-significant agreement between UL and LL measures to identify older adults with low SMS ($\kappa$ = 0.046; 95% CI 0.093–0.185; p = 0.352). There was a moderate agreement between UL and LL measures to identify older

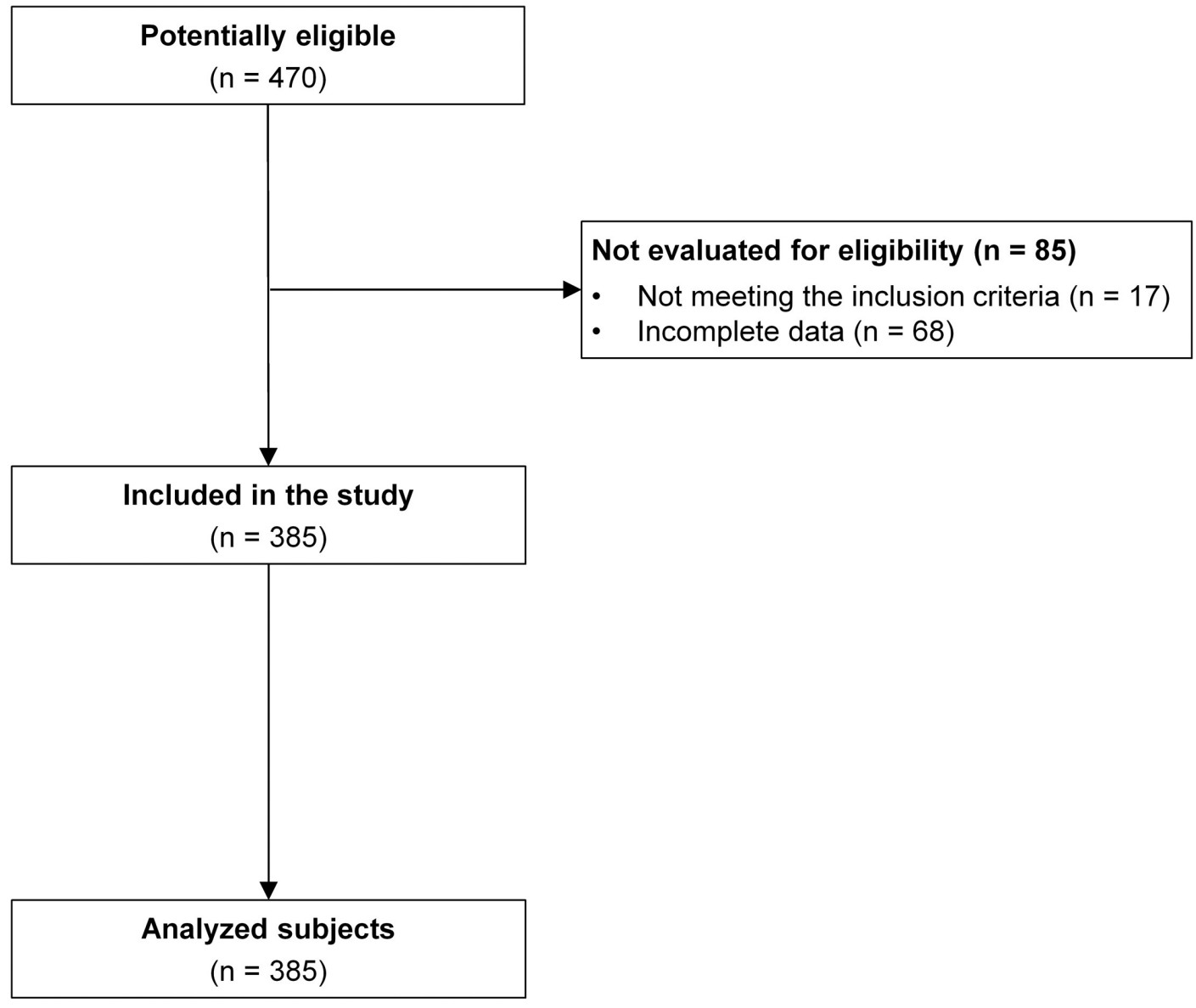

**Fig 1. Study flowchart.**

adults with low SMM (κ = 0.473; 95% CI 0.371–0.574; p = 0.001). There was a fair agreement between UL and LL measures to identify older adults with low SMQ (κ = 0.206; 95% CI 0.082 to 0.330; p = 0.005).

Tables 4 and 5 shows the agreement analysis between UL and LL measures to identify older males and females with low SMS, SMM, and SMQ. There was a slight and non-significant agreement between UL and LL measures to identify low SMS in older males (κ = 0.183; 95% CI -0.071–0.436; p = 0.059). There was a fair agreement between UL and LL measures to identify low SMM in older males (κ = 0.376; 95% CI -0.071–0.436; p = 0.001) and a fair agreement to identify low SMQ in older males (κ = 0.448; 95% CI 0.223 to 0.673; p = 0.001). There was a poor and non-significant agreement between UL and LL measures to identify low SMS in older females (κ = -0.001; 95% CI -0.166–0.164; p = 0.987). There was moderate agreement

**Table 1. Characteristics of the participants (n = 385).**

| | Total | Males | Females |
|---|---|---|---|
| N (%) | 385 (100) | 106 (27.6) | 279 (72.4) |
| Age (years) | 66.1 ± 4.5 | 66.0 ± 4.4 | 66.1 ± 4.6 |
| Height (cm) | 157.0 ± 8.43 | 166.3 ± 6.72 | 153.4 ± 5.96 |
| Body weight (kg) | 71.2 ± 13.83 | 77.4 ± 13.47 | 68.8 ± 13.27 |
| Body mass index (kg/m$^2$) | 28.8 ± 4.86 | 27.9 ± 4.31 | 29.1 ± 5.01 |
| Total skeletal muscle mass (kg) | 40.7 ± 8.34 | 50.3 ± 6.93 | 37.1 ± 5.47 |
| Arm skeletal muscle mass (kg) | 4.5 ± 1.30 | 6.08 ± 1.10 | 3.98 ± 0.82 |
| Leg skeletal muscle mass (kg) | 14.1 ± 3.05 | 17.2 ± 2.69 | 12.9 ± 2.25 |
| Handgrip strength test (kg) | 29.0 ± 8.09 | 39.1 ± 6.41 | 25.2 ± 4.57 |
| 30-s chair stand test (rep) | 13.3 ± 3.80 | 14.9 ± 4.40 | 12.6 ± 3.30 |
| Upper limb skeletal muscle quality (kgf/kg) | 0.71 ± 0.12 | 0.78 ± 0.12 | 0.68 ± 0.11 |
| Lower limb skeletal muscle quality (rep/kg) | 0.33 ± 0.11 | 0.30 ± 0.11 | 0.35 ± 0.11 |

Data are expressed as mean ± standard deviation. Rep = repetition.

between UL and LL measures to identify low SMM in older females ($\kappa$ = 0.507; 95% CI 0.384–0.629; p = 0.001) and a fair agreement to identify low SMQ in older females ($\kappa$ = 0.126; 95% CI -0.019–0.271; p = 0.001).

## 4 Discussion

To the best of our knowledge, this is the first study which has investigated the agreement between UL and LL measures to identify older adults with low SMS, SMM and SMQ. The main findings indicate that: i) there was slight and non-significant agreement between UL and LL measures to identify older adults with low SMS; ii) there was a moderate and fair agreement between UL and LL measures to identify older adults with low SMM and SMQ, respectively.

Despite the HGS and 30-s CST being well-recognized tests to measure SMS, we did not observe a significant agreement between them to identify older adults with low SMS. The decline of SMS occurs in different magnitudes over aging in UL and LL [1–4, 31,32]. Frontera et al. [33] showed a decline of 1.4 and 2.5% per year in UL- and LL-SMS, respectively. Other studies have observed a higher magnitude of difference (i.e. a decline of 1.4 and 5.4% per year in UL- and LL-SMS, respectively) [19, 34]. It seems clear that the LL-SMS declines to a greater magnitude with aging than the UL-SMS, which can partially explain our findings. Recently, Yeung et al [35]. investigated the agreement between the HGS and knee extension strength

**Table 2. Cut-offs for upper and lower limb measures to identify older adults with low skeletal muscle strength, muscle mass, and muscle quality according to sex and age group.**

| | Males | | Females | |
|---|---|---|---|---|
| | 60–69 yr | 70–80 yr | 60–69 yr | 70–80 yr |
| Handgrip strength test (kg) | 36.0 | 34.0 | 23.0 | 21.0 |
| 30-s chair stand test (rep) | 12.0 | 12.0 | 11.0 | 10.0 |
| Arm skeletal muscle mass (kg) | 6.0 | 5.0 | 3.5 | 3.0 |
| Leg skeletal muscle mass (kg) | 15.0 | 15.0 | 12.0 | 11.0 |
| Upper limb skeletal muscle quality (kgf/kg) | 0.70 | 0.61 | 0.72 | 0.59 |
| Lower limb skeletal muscle quality (rep/kg) | 0.24 | 0.22 | 0.28 | 0.27 |

Cut-offs were defined as values below 25[th] percentile for sex and age group. Rep = repetition.

**Table 3. Agreement between upper and lower limb measures to identify older adults with low skeletal muscle strength, muscle mass and muscle quality.**

| Skeletal muscle strength | Low UL-SMS | Normal UL-SMS | Kappa | 95% CI | P |
|---|---|---|---|---|---|
| Low LL-SMS | 20 (27.8%) | 52 (72.2%) | 0.046 | 0.093 to 0.185 | 0.352 |
| Normal LL-SMS | 71 (22.7%) | 242 (77.3%) | | | |
| **Skeletal muscle mass** | Low UL-SMM | Normal UL-SMM | | | |
| Low LL-SMM | 67 (60.4%) | 44 (39.6%) | 0.473 | 0.371 to 0.574 | 0.001 |
| Normal LL-SMM | 38 (13.9%) | 236 (86.1%) | | | |
| **Skeletal muscle quality** | Low UL-MQ | Normal UL-SMQ | | | |
| Low LL-SMQ | 37 (39.8%) | 56 (60.2%) | 0.206 | 0.082 to 0.330 | 0.005 |
| Normal LL-SMQ | 56 (19.2%) | 236 (80.8%) | | | |

UL = upper limb; LL = lower limb; SMS = skeletal muscle strength; SMM = skeletal muscle mass; SMQ = skeletal muscle quality; CI: confidence interval.

performance in individuals from different age and health-status groups, in which they observed a low correlation between HGS and knee extension strength in healthy older adults and a moderate correlation in geriatric outpatients and older adults post-hip fracture. The authors found poor to moderate intraclass correlation coefficients between the tests. At an individual level, Bland-Altman plots indicated that the agreement between HGS and knee extension strength was lower among healthy older adults compared to geriatric outpatients and older adults post-hip fracture. Taken together, even using a different test to assess LL-SMS and other statistical approaches, the results from Yeung et al. [35] seem to be in accordance with our findings regarding the limited agreement between UL and LL measures to assess SMS in healthy older adults.

Another reason which can explain the non-significant agreement between the HGS and 30-s CST to identify older adults with low SMS is the characteristics of these tests. Although HGS and 30-s CST are valid proxies of SMS [36], the HGS measures the maximal isometric contraction, while the 30-s CST assesses the performance on a functional task, which seems to involve other fitness-related components in addition to SMS [23]. In accordance with our findings, Johansson et al. [20] found a poor agreement between HGS and 5-repetition CST to identify older adults at high risk for ($\kappa = 0.07$) and having established sarcopenia ($\kappa = 0.18$). Moreover, only 1.3% and 4.4% of the older adults were identified as having a high risk for or having established sarcopenia by both HGS and 5-repetition CST, respectively.

A significant agreement between UL and LL measures to identify low SMM in older adults was observed in the present study. Different from SMS, the decline of SMM seems to occur in a similar magnitude over aging in UL and LL in some studies [17, 37, 38], which may explain

**Table 4. Agreement between upper and lower limb measures to identify older males with low skeletal muscle strength, muscle mass and muscle quality.**

| Skeletal muscle strength | Low UL-SMS | Normal UL-SMS | Kappa | 95% CI | P |
|---|---|---|---|---|---|
| Low LL-SMS | 08 (33.3%) | 16 (66.7%) | 0.183 | -0.071 to 0.436 | 0.059 |
| Normal LL-SMS | 13 (15.9%) | 69 (84.1%) | | | |
| **Skeletal muscle mass** | Low UL-SMM | Normal UL-SMM | | | |
| Low LL-SMM | 24 (51.2%) | 18 (48.8%) | 0.376 | 0.191 to 0.561 | 0.001 |
| Normal LL-SMM | 13 (3.2%) | 51 (96.8%) | | | |
| **Skeletal muscle quality** | Low UL-MQ | Normal UL-SMQ | | | |
| Low LL-SMQ | 12 (63.2%) | 07 (36.8%) | 0.448 | 0.223 to 0.673 | 0.001 |
| Normal LL-SMQ | 12 (13.8%) | 75 (86.2%) | | | |

UL = upper limb; LL = lower limb; SMS = skeletal muscle strength; SMM = skeletal muscle mass; SMQ = skeletal muscle quality; CI: confidence interval.

**Table 5. Agreement between upper and lower limb measures to identify older females with low skeletal muscle strength, muscle mass and muscle quality.**

| Skeletal muscle strength | Low UL-SMS | Normal UL-SMS | Kappa | 95% CI | P |
|---|---|---|---|---|---|
| Low LL-SMS | 12 (25.0%) | 36 (75.0%) | -0.001 | -0.166 to 0.164 | 0.987 |
| Normal LL-SMS | 58 (25.1%) | 173 (74.9%) | | | |
| **Skeletal muscle mass** | Low UL-SMM | Normal UL-SMM | | | |
| Low LL-SMM | 43 (62.3%) | 26 (37.7%) | 0.507 | 0.384 to 0.629 | 0.001 |
| Normal LL-SMM | 25 (11.9%) | 185 (88.1%) | | | |
| **Skeletal muscle quality** | Low UL-MQ | Normal UL-SMQ | | | |
| Low LL-SMQ | 25 (33.8%) | 49 (66.2%) | 0.126 | -0.019–0.271 | 0.001 |
| Normal LL-SMQ | 44 (21.5%) | 161 (78.5%) | | | |

UL = upper limb; LL = lower limb; SMS = skeletal muscle strength; SMM = skeletal muscle mass; SMQ = skeletal muscle quality; CI: confidence interval.

the agreement between UL and LL measures observed in this study. On the other hand, some studies show a reduction in different magnitudes between UL and LL, depending on how and where we measure [39, 40]. Despite this, the magnitude of this agreement was moderate. It is reasonable to think that other factors can explain the moderate agreement between UL and LL measures to identify low SMM in older adults. The DXA technique assesses the LST, which includes ~55% of SMM [24, 25]. The additional components of LST (body water, carbohydrates, nonfat lipids, and soft tissue mineral) can be different between UL and LL, which could also explain the moderate agreement observed between UL- and LL-SMM [41]. In addition, ~75% of SMM are concentrated in the LL and the rest is distributed in the trunk and in the UL [24]. This aspect can partially explain the moderate agreement between UL and LL measures to identify low SMM in older adults. Given that 60.4% of the older adults were identified as having low SMM by both arm and leg measures and 39.6% were identified as having low SMM in only one of these measures, it seems reasonable to assume that the UL and LL measures might induce different clinical interpretations regarding identification of low SMM in older adults.

Regarding the SMQ, which is an index derived from the SMS and SMM [6, 42], a significant but fair agreement was observed between UL and LL measures to identify low SMQ in older adults. Only 39.8% of the older adults were identified as having low SMQ by both UL and LL measures. We believe that this finding may be explained by the dissimilar performance of the older adults in the UL and LL tests to assess SMS. It should be noted that the UL- and LL-SMQ indexes are dissimilar in their nature due to the different characteristics of the SMS tests. The UL-SMQ index refers to a maximal isometric SMS normalized by SMM, which is commonly reported in the literature [6]. The LL-SMQ index refers to maximal performance on an LL functional task normalized by SMM, in which its ability seems to not be exclusively dependent of the maximal dynamic LL-SMS. Although the 30-s CST shows a high correlation with one-repetition maximum test on the leg press (older females: r = 0.71; older males: r = 0.78) [23], which is a multi-joint exercise involving the hips, knees, and ankles, it seems reasonable to assume the 30-s CST performance requires additional fitness-related components in addition to maximal dynamic SMS, such as dynamic balance, coordination, and power. We believe that the above-mentioned aspects may explain the fair agreement between UL and LL measures to identify low SMQ in older adults.

From a clinical perspective, our findings might be useful to rethink the recommendation of the interchangeable use of the HGS and CST in the EWGSOP2 [7] practical algorithm for dynapenia and sarcopenia case-finding, diagnosis and severity, mainly due the limited agreement between these UL and LL tests to identify low SMS and SMQ in older adults. Based on

our findings, using the HGS an older adult can be classified as 'normal SMS' and nonsarcopenic while using CST his/her classification can be dynapenia (low SMS) or even sarcopenia. The opposite scenario is also possible; i.e. 'normal SMS' and nonsarcopenic using CST and dynapenia or sarcopenia using HGS. Thus, misinterpretation regarding the clinical identification of dynapenia and sarcopenia can occur, which can favor unappropriated interventions delivered for these individuals.

Despite our novel and interesting findings, this study has limitations which should be mentioned. First, although HGS and 30-s CST are well recommended to assess SMS in older adults by clinical guidelines, including the EWGSOP2 [7], these tests have different characteristics which may have influenced our findings. Future studies could consider investigating the agreement between UL and LL measures to identify older adults with low SMS using tests with similar characteristics. Second, the cut-offs to determine low SMS, SMM and SMQ were defined according to sex and 10-year age groups due to a low number of participants aged 75–80 years. Other studies suggest determining neuromuscular characteristics and fitness-related performance cut-offs for 5-year age groups [23, 27]. Third, our study included older adults aged 60–80 years. Therefore, our findings should be interpreted with caution and they are not transferable to older adults aged > 80 years. Fourth, we recruited community-dwelling older adults by diverse advertisement methods, but we do not rule out the possibility of some selection bias due to the need for transportation to the research laboratory. This aspect might have limited the participation of older adults with poor mobility and other age-related conditions such as sarcopenia and frailty. Fifth, as previously described, the DXA technique assesses the LST, which includes SMM and other body composition components [24, 25]. Although LST is highly correlated with SMM assessed by magnetic resonance imaging and computerized tomography imaging [24] and is a well-recognized proxy of SMM, the DXA technique does not provide a specific evaluation of SMM.

## 5 Conclusion

The agreement between UL and LL measures to identify low SMS, SMM and SMQ in older adults is limited, which might generate different clinical interpretations for diagnosing sarcopenia cases. In order to establish better implications of our findings, it seems important to identify which neuromuscular UL or LL measure (SMS, SMM and SMQ) is more associated with adverse health-related outcomes in older adults. Future studies to address the above-mentioned question are important.

## Author Contributions

**Conceptualization:** Charles Phillipe de Lucena Alves, Marcyo Câmara, Geovani Araújo Dantas Macêdo, Yuri Alberto Freire, Raíssa de Melo Silva, Ronildo Paulo-Pereira, Luiz Fernando Farias-Junior, Ana Paula Trussardi Fayh, Arnaldo Luis Mortatti, Eduardo Caldas Costa.

**Formal analysis:** Charles Phillipe de Lucena Alves.

**Investigation:** Charles Phillipe de Lucena Alves, Marcyo Câmara, Geovani Araújo Dantas Macêdo, Yuri Alberto Freire, Raíssa de Melo Silva, Ronildo Paulo-Pereira, Luiz Fernando Farias-Junior, Ana Paula Trussardi Fayh, Arnaldo Luis Mortatti, Eduardo Caldas Costa.

**Methodology:** Charles Phillipe de Lucena Alves, Marcyo Câmara, Geovani Araújo Dantas Macêdo, Yuri Alberto Freire, Raíssa de Melo Silva, Ronildo Paulo-Pereira, Luiz Fernando Farias-Junior, Ana Paula Trussardi Fayh, Arnaldo Luis Mortatti, Eduardo Caldas Costa.

**Project administration:** Charles Phillipe de Lucena Alves, Arnaldo Luis Mortatti, Eduardo Caldas Costa.

**Supervision:** Eduardo Caldas Costa.

**Writing – original draft:** Charles Phillipe de Lucena Alves, Marcyo Câmara, Geovani Araújo Dantas Macêdo, Yuri Alberto Freire, Raíssa de Melo Silva, Ronildo Paulo-Pereira, Luiz Fernando Farias-Junior, Ana Paula Trussardi Fayh, Arnaldo Luis Mortatti, Eduardo Caldas Costa.

**Writing – review & editing:** Charles Phillipe de Lucena Alves, Marcyo Câmara, Geovani Araújo Dantas Macêdo, Yuri Alberto Freire, Raíssa de Melo Silva, Ronildo Paulo-Pereira, Luiz Fernando Farias-Junior, Ana Paula Trussardi Fayh, Arnaldo Luis Mortatti, Eduardo Caldas Costa.

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
