## [Decision Letter · Decision Letter 0]

28 Oct 2021

PONE-D-21-18967Agreement between upper and lower limb measures to identify older adults with low skeletal muscle strength, muscle mass and muscle qualityPLOS ONE

Dear Dr. Costa,

Thank you for submitting your manuscript to PLOS ONE. After careful consideration, we feel that it has merit but does not fully meet PLOS ONE’s publication criteria as it currently stands. Therefore, we invite you to submit a revised version of the manuscript that addresses the points raised during the review process.

Please submit your revised manuscript by December 12, 2021. If you will need more time than this to complete your revisions, please reply to this message or contact the journal office at plosone@plos.org. Please include the following items when submitting your revised manuscript:A rebuttal letter that responds to each point raised by the academic editor and reviewer(s). You should upload this letter as a separate file labeled 'Response to Reviewers'.A marked-up copy of your manuscript that highlights changes made to the original version. You should upload this as a separate file labeled 'Revised Manuscript with Track Changes'.An unmarked version of your revised paper without tracked changes. You should upload this as a separate file labeled 'Manuscript'.

We look forward to receiving your revised manuscript.

Kind regards,

Alessandra Coin

Academic Editor

PLOS ONE

Additional Editor Comments (if provided):

Dear author,

I'm very sorry for the delay but it was incredibly difficult to find reviewers, probably for the post-pandemic period and not for the topic of your manuscript which is very interesting.

For this reason, to include the process and to answer you I decided to revise your study by myself as second reviewer.

In this study authors assess the agreement between upper and lower limb indexes of sarcopenia in 385 subjects aged 60-80. They found that the agreement between upper and lower limbs strength was non significant, that between upper and lower limbs muscle quality was fair, whereas the agreement was moderate considering upper and lower limbs muscle mass.

The premises and the aim of the study are interesting given the importance of detecting persons at risk for/affected by sarcopenia.

In my opinion the major limitation of the study is the fact that lower limbs muscle strength was measured by means of th 30-s chair stand test rather than with a dynamometer. As Authors state 30-s chair stand test assesses the performance on a functional test, being not only an indicator of muscle strength.

It would be interesting to add in Results section and/or in Table 1 other general characteristics of the sample, as indicators of physical performance, subjects' level of physical activity.

Finally, the Authors should discuss the reason why the agreement between upper and lower limbs muscle strength and quality is so relevant.

I suggest a punctuation check (see Results section, lines 167-168: change 35,3% in 35.3%; and 32,99% in 32.99%).

Thanks for the attention, sorry again for the long period.

Best regards

Journal Requirements:

3. Please change "female” or "male" to "woman” or "man" as appropriate, when used as a noun (see for instance https://apastyle.apa.org/style-grammar-guidelines/bias-free-language/gender).

Please improve statistical reporting and ensure decimal values are represented by periods instead of commas. Our statistical reporting guidelines are available at https://journals.plos.org/plosone/s/submission-guidelines#loc-statistical-reporting.

Reviewers' comments:

Reviewer's Responses to Questions

**Comments to the Author**

1. Is the manuscript technically sound, and do the data support the conclusions?

Reviewer #1: Yes

2. Has the statistical analysis been performed appropriately and rigorously? 

Reviewer #1: Yes

3. Have the authors made all data underlying the findings in their manuscript fully available?

Reviewer #1: Yes

4. Is the manuscript presented in an intelligible fashion and written in standard English?

Reviewer #1: Yes

5. Review Comments to the Author

Reviewer #1: The authors assess the agreement between upper and lower limb measures of muscle mass, strength and quality in older adults.

The topic is interesting, the sample size is adequate and statistical analysis is well conducted. The paper is clearly written.

6. PLOS authors have the option to publish the peer review history of their article (what does this mean?). If published, this will include your full peer review and any attached files.

Reviewer #1: No

---

## [Author Response · Author response to Decision Letter 0]

2 Dec 2021

RE: Response letter of the manuscript ID PONE-D-21-18967, entitled ‘Agreement between upper and lower limb measures to identify older adults with low skeletal muscle strength, muscle mass and muscle quality’, for review.

Dear Dr. Emily Chenette (Editor-In-Chief)

We appreciate the consideration of our manuscript for review in the PLOS ONE, which is a cutting-edge journal in the field of biomedical literature. We have read with great interest the concerns addressed by the reviewers and we are thankful for the time they have spent reviewing our manuscript. All suggestions were valuable to the improvement of the manuscript and they made the authors rethink many aspects of the tackled issues. Therefore, that said, we listed and respond to all concerns and issues raised by the reviewers and editors in our manuscript. The changes are listed below and highlight in the manuscript: 

Editor and Reviewer Comments

Editor comments

Comment 1: The premises and the aim of the study are interesting given the importance of detecting persons at risk for/affected by sarcopenia.

Response: Thank you for the feedback and comments. We are grateful for the time you spent reviewing our manuscript.

Comment 2: In my opinion the major limitation of the study is the fact that lower limbs muscle strength was measured by means of the 30-s chair stand test rather than with a dynamometer. As Authors state 30-s chair stand test assesses the performance on a functional test, being not only an indicator of muscle strength.

Response: Thank you for the feedback and the opportunity to clarify this point. Our rationale was based on the guideline from the European Working Group on Sarcopenia in Older People (EWGSOP2), which recommends assessing skeletal muscle strength by using interchangeable tests; i.e. handgrip strength test or chair stand test. This guideline, which is the most used in clinical practice worldwide and designed “to increase consistency of research design, clinical diagnoses and ultimately, care for people with sarcopenia” states that:

“The chair stand test (also called chair rise test) can be used as a proxy for strength of leg muscles (quadriceps muscle group). The chair stand test measures the amount of time needed for a patient to rise five times from a seated position without using his or her arms; the timed chair stand test is a variation that counts how many times a patient can rise and sit in the chair over a 30-second interval [64, 67, 68]. Since the chair stand test requires both strength and endurance, this test is a qualified but convenient measure of strength”. (pg. 20). 

Therefore, although we fully agree that handgrip strength test and chair stand test are different, their use are recommended by the most used sarcopenia guideline in clinical practice. Therefore, our findings might be useful to rethink the interchangeable use of these different tests in the practical algorithm for sarcopenia case-finding, diagnosis and severity.

Comment 3: It would be interesting to add in Results section and/or in Table 1 other general characteristics of the sample, as indicators of physical performance, subjects' level of physical activity.

Response: Thank for this feedback. Information about participants’ physical activity level has been added accordingly. 

Methods

2.7 Physical activity

Physical activity level was determined by the Brazilian version of the Minnesota Leisure Time Activities Questionnaire for older adults. The physical activities were classified as light, moderate or vigorous considering the absolute intensity (metabolic equivalents; METs) for each specific age (40-64 years; ≥ 65 years), based on the American College of Sports Medicine. Participants who performed ≥600 MET/min/wk of moderate-vigorous physical activities were considered as ‘active’, while those who performed < 600 MET/min/wk were considered as ‘inactive’.

Results

Additionally, 58.4% (n = 225) were physically active and 41.6% (n = 160) were physically inactive.

Comment 4: Finally, the Authors should discuss the reason why the agreement between upper and lower limbs muscle strength and quality is so relevant.

Response: Thank you for the feedback. In the discussion section, this information has been included accordingly.

From a clinical perspective, our findings might be useful to rethink the recommendation of the interchangeable use of the HGS and CST in the EWGSOP2 (7) practical algorithm for dynapenia and sarcopenia case-finding, diagnosis and severity, mainly due the limited agreement between these UL and LL tests to identify low SMS and SMQ in older adults. Based on our findings, using the HGS an older adult can be classified as ‘normal SMS’ and nonsarcopenic while using CST his/her classification can be dynapenia (low SMS) or even sarcopenia. The opposite scenario is also possible; i.e. ‘normal SMS’ and nonsarcopenic using CST and dynapenia or sarcopenia using HGS. Thus, misinterpretation regarding the clinical identification of dynapenia and sarcopenia can occur, which can favor unappropriated interventions delivered for these individuals.

Comment 5: I suggest a punctuation check (see Results section, lines 167-168: change 35,3% in 35.3%; and 32,99% in 32.99%).

Response: Thank you for the feedback. The changes have been made accordingly.

Reviewer #1: The authors assess the agreement between upper and lower limb measures of muscle mass, strength and quality in older adults. The topic is interesting, the sample size is adequate and statistical analysis is well conducted. The paper is clearly written.

Response: Thank you for the feedback and comments. We are grateful for the time you spent reviewing our manuscript.

---

## [Editor Report · Decision Letter 1]

5 Jan 2022

Agreement between upper and lower limb measures to identify older adults with low skeletal muscle strength, muscle mass and muscle quality

PONE-D-21-18967R1

Dear Dr. Eduardo Caldas Costa,

We’re pleased to inform you that your manuscript has been judged scientifically suitable for publication and will be formally accepted for publication once it meets all outstanding technical requirements.

Kind regards,

Alessandra Coin

Academic Editor

PLOS ONE

---

## [Editor Report · Acceptance letter]

10 Jan 2022

PONE-D-21-18967R1 

Agreement between upper and lower limb measures to identify older adults with low skeletal muscle strength, muscle mass and muscle quality 

Dear Dr. Costa:

I'm pleased to inform you that your manuscript has been deemed suitable for publication in PLOS ONE. Congratulations! Your manuscript is now with our production department. 

Kind regards, 

on behalf of

Dr. Alessandra Coin 

Academic Editor

PLOS ONE